# Polylactic Acid Film Coated with Electrospun Gelatin/Chitosan Nanofibers Containing Betel Leaf Ethanolic Extract: Properties, Bioactivities, and Use for Shelf-Life Extension of Tilapia Slices

**DOI:** 10.3390/molecules27185877

**Published:** 2022-09-10

**Authors:** Mohamed Tagrida, Saqib Gulzar, Krisana Nilsuwan, Thummanoon Prodpran, Bin Zhang, Soottawat Benjakul

**Affiliations:** 1International Center of Excellence in Seafood Science and Innovation, Faculty of Agro-Industry, Prince of Songkla University, Hat Yai 90110, Songkhla, Thailand; 2Center of Excellence in Bio-based Materials and Packaging Innovation, Faculty of Agro-Industry, Prince of Songkla University, Hat Yai 90110, Songkhla, Thailand; 3College of Food and Pharmacy, Zhejiang Ocean University, Zhoushan 316000, China

**Keywords:** gelatin, chitosan, betel leaf extract, PLA films, electrospinning, bioactivities

## Abstract

Gelatin/chitosan solutions incorporated with betel leaf ethanolic extract (BLEE) at varying concentrations were electrospun on polylactic acid (PLA) films. Nanofibers with different morphologies, as indicated by scanning electron microscopy (SEM), were formed after solutions of gelatin/chitosan with and without BLEE were electrospun on PLA films at a constant voltage (25 kV) and a feed rate of 0.4 mL/h. Beaded gelatin/chitosan nanofibers (GC/NF) were found, particularly when high concentrations of BLEE were encapsulated. PLA films coated with GC/NF, and with BLEE added, showed antioxidant and antibacterial activities, which were augmented by increasing BLEE concentrations. Lower water vapor permeability and enhanced mechanical properties were achieved for GC/NF-coated PLA film (*p* < 0.05). Microbial growth and lipid oxidation of Nile tilapia slices packaged in PLA film coated with GC/NF containing 2% BLEE were more retarded than those packaged in low-density polyethylene (LDPE) bags over refrigerated storage of 12 days. Based on microbial limits, the shelf-life was escalated to 9 days, while the control had a shelf-life of 3 days. Therefore, such a novel film/bag could be a promising active packaging for foods.

## 1. Introduction

Generally, most food packaging materials are produced from non-biodegradable petrochemical plastics, which are the major cause of environmental pollution. There is an urgent need for natural alternatives, which are environmentally friendly; these are essential for implementation in food packaging [1]. Recently, food packaging design is not limited to packaging and protection, but also includes “active packaging”, which is the packaging that interacts positively with foods and promotes benefits, in particular for retarding microbial growth and preventing oxidation [2].

Several natural polymers are used for preparation of biodegradable active packaging to reduce the usage of their non-degradable counterparts [3]. Polylactic acid (PLA) is one such polymer, and is the fermented product from some types of agricultural crops or waste. It is classified as being “Generally Recognized as Safe (GRAS)” [4]. Due to its excellent mechanical and physical properties, PLA was introduced to replace common petrochemicals used in food packaging [5]. Nevertheless, several limitations, such as low thermal stability and gas barrier dysfunctionality, may hinder the use of PLA at a commercial scale. In an attempt to tackle such limitations, different approaches were implemented. The lamination of other biopolymers with better barrier abilities on the PLA film is a promising technique; in this technique, the barrier property was improved, thus showing better protection toward packaged items. Therefore, the physical and mechanical properties of PLA film could be enhanced by mixing with other biopolymers [6].

Gelatin and chitosan are used as biopolymers in food packaging because of their biocompatibility, biodegradability, and other eminent characteristics [7]. Laminating PLA films with gelatin and chitosan can overcome the limitations of its barrier abilities and stability. Bilayer and/or multilayer films made from such biodegradable polymers are considered an innovative approach [8]. Additionally, the production of nanofibers with a good nanostructure and mechanical property requires a mixture of polymers, such as gelatin or chitosan, because a single polymer cannot produce nanofibers of adequate quality [9].

Materials at the nanometric scale, such as nanofibers, can not only increase the contacted surface area, but can also be loaded with bioactive agents, for application as active packaging [9,10]. Bioactive compounds, such as plant extracts or essential oils having antioxidant and antimicrobial activities, were widely used in active packaging [11]. Betel, grown in Southeast Asian countries, is a leafy plant used in traditional medicine and as a food, due to its bioactivities. Betel leaf ethanolic extract (BLEE), dechlorophyllized using the sedimentation process, contained numerous phenolic compounds, such as isovitexin, vitexin 4′-*O*-galactoside, and kaempferol derivatives, possessing immense bioactivities [12]. BLEE could, therefore, be used to develop biodegradable active packaging, especially via embedding in nanofibers.

Electrospinning gained attention because it can produce different bioactive packaging systems, particularly by the loading and controlled release of natural compounds [13]. Electrospinning can generate a mat of polymer nanofibers that can be loaded with bioactive compounds [14]. Additionally, some techniques, such as nozzleless electrospinning, are capable of producing nanofibers on a large or commercial scale [15]. The high surface area-to-volume ratio of nanofibers with elevated porosity guarantees the sustained release of the bioactive compounds [16]. This technology is based on high electric fields applied to viscoelastic polymer-based solutions containing the bioactive compounds. The resultant nonwoven nanofibers containing the bioactive compounds can be applied as coatings on films to form a novel active packaging system [17].

The purpose of the current work was to produce PLA films coated with electrospun gelatin/chitosan nanofibers loaded with BLEE. The mechanical, physical, and bioactive properties of the resulting films were examined. Additionally, the quality changes of tilapia slices stored in bags made from the selected PLA film coated with gelatin/chitosan electrospun fiber (GC/NF) containing BLEE was also monitored during refrigerated storage.

## 2. Results and Discussion

### 2.1. Properties of PLA Films Coated with Electrospun GC/NF with, and without, BLEE at Different Concentrations

#### 2.1.1. Morphology

SEM images of all the films are depicted in Figure 1A. Smooth homogenous and bead-free nanofibers, with diameters ranging from 35 to 116 nm, were coated on PLA films, specifically, those with or without 0.5% BLEE (Figure 1A,B). However, when the BLEE concentration was augmented, nanofibers with diameters ranging from 56 to 183 nm were observed and beads were also found (Figure 1C,D). Bead formation could be attributed to the decreased viscosity in the nanofiber-forming solutions, plausibly caused by several compounds in BLEE, which impeded the entanglement of nanofibers [12]. Viscosity is a crucial factor determining the structure and diameter of the electrospun nanofibers. Solutions with elevated viscosity have higher viscoelastic forces which can resist axial stretching, resulting in diameter increment and less bead formation [18]. On the other hand, solutions with lower viscosity exhibited a higher surface tension in the charged jet, thus overpowering the viscoelastic forces. This led to the phenomenon termed Rayleigh–Taylor instability, which occurs when solutions with different fluidities become unstable at their interface due to the penetration of the heavy fluids into the lighter fluids [19]. The aforementioned phenomenon resulted in the beaded nanofibers of lower diameter. The viscosity of the electrospinning solutions was 48.44 ± 0.91, 47.13 ± 1.04, 44.86 ± 1.12, and 43.01 ± 1.32 cP for GC/NF-0%BLEE, GC/NF-0.5%BLEE, GC/NF-1%BLEE, and GC/NF-2%BLEE, respectively. Additionally, the conductivity of the GC/NF-forming solutions also determined the morphology of the resulting nanofiber mat. The solutions with higher BLEE concentrations had lower conductivity, leading to the formation of nanofiber mats of decreased diameters. This is in line with Vafania, Fathi, and Soleimanian-Zad [20] who documented that electrospinning solutions with lower conductivity, due to augmented thyme essential-oil concentration, had a lower nanofiber diameter. In the current study, the conductivity of the electrospinning solutions was 553.17 ± 3.22, 542.77 ± 4.62, 519.76 ± 4.36, and 509.41 ± 2.19 μS/cm for GC/NF-0%BLEE, GC/NF-0.5%BLEE, GC/NF-1%BLEE, and GC/NF-2%BLEE, respectively. Some hydrophobic compounds in the extract might lower conductivity of the GC/NF forming solutions. These results indicate that BLEE concentration affected the nanofiber diameter size and morphology by impacting the viscosity and conductivity of the electrospinning solutions.

#### 2.1.2. Thickness

Thickness is another factor influencing the mechanical and barrier properties of the produced films. PLA films coated with GC/NF with and without BLEE had an average thickness ranging between 0.115 and 0.121 mm (Table 1), which was generally greater than the control PLA film (0.106 mm) (*p* < 0.05). Films containing BLEE at elevated concentrations (1 and 2%) were thicker than those with 0.5%, or without any BLEE (*p* < 0.05). However, no differences in thickness were detected among the films containing BLEE at different levels (*p* > 0.05). The fibrous network could augment the thickness of resulting films. The results indicate that embedding of BLEE and its concentration, especially at 1 and 2%, increased the thickness of the resulting films. Similarly, increased thickness was attributed to the embedding of pomegranate peel extract and sodium dehydro-acetate, which contain both soluble and insoluble matter [13].

#### 2.1.3. Mechanical Properties

Mechanical strength is vital to maintain the integrity of a film when it is used as food packaging, and during transportation and storage. Mechanical properties are expressed as TS and EAB (Table 1). Overall, the TS and EAB of PLA films coated with GC/NF containing BLEE were higher than those without the coating (*p* < 0.05), indicating a significant improvement in mechanical properties. However, no difference in TS was found between PLA films and that coated with GC/NF containing no BLEE (*p* > 0.05). GC/NF coating on PLA surface plausibly augmented TS, in which nanofiber entanglement could strengthen the resulting film. Ebrahimi, Fathi, and Kadivar [15] found that nanofibers made from chitosan/gelatin and coated on a gelatin-film surface improved the TS of the film by acting like scaffolds in the film structure. The stretch ability of GC/NF-coated PLA films was enhanced when BLEE concentrations above 0.5% were added into the GC/NF, compared with the uncoated PLA films, as ascertained by an increased EAB (*p* < 0.05). Coating with GC/NF probably enhanced the elasticity of the PLA film by rearranging the nanofiber network, leading to the increased stretching [21]. Additionally, the presence of hydrogen bonding mediated by chitosan might enhance the mechanical properties of the PLA films, in which chitosan could act as an H-donor, thus strengthening the nanofiber network [22]. The addition of BLEE enhanced the EAB, probably via increasing hydrogen bonding due to the presence of several -OH groups in the BLEE. Overall, the enhancement of the mechanical properties of the PLA films coated with GC/NF containing BLEE was observed when compared with the uncoated PLA films, signifying the valuable effect of electrospinning as a promising means of producing food packaging with improved mechanical properties.

#### 2.1.4. Water Vapor Permeability (WVP)

WVP is the measure for the water vapor migration rate through the surface of a film. Usually, in food packaging, films with a low WVP are favored as they prevent the migration of water vapor—as moisture from the environment—to packaged food products [23]. The WVP of uncoated PLA films, and those with the GC/NF coating with and without BLEE, is shown in Table 1. WVP was observed to be reduced when coated with GC/NF and when the BLEE concentration was increased, especially at 1 and 2% BLEE (*p* < 0.05), suggesting that coating nanofibers with high levels of BLEE augmented the resistance of the film to the migration of water vapor. Types of polymers and additives generally affect the barrier properties of the films. PLA is a hydrophobic polymer with moderate WVP, and it can suppress the easy migration of water molecules [24]. Cross-linking between gelatin and chitosan created a dense nanofiber mat that was able to prevent water vapor permeability [18]. BLEE also increased the cross-linking between the gelatin and chitosan molecules, leading to the decrease in the hydrophilic functional groups and lower water migration [25]. In addition, BLEE contains many hydrophobic compounds, such as eugenol, estragole, chavicol, linalool, etc., that might repel the water molecules [26]. The presence of the GC/NF layer on the surface of the PLA film reduced the WVP. This reduction was more prominent when the BLEE concentration was above 0.5% (*p* < 0.05). Fabra, López-Rubio, and Lagaron [27] documented that coating a zein/gluten bilayer film with electrospun nanofibers reduced the WVP effectively. Therefore, PLA film coated with GC/NF containing BLEE could improve the water vapor barrier property of the resulting films.

#### 2.1.5. Oxygen Permeability (OP)

A crucial property in food packaging materials is their oxygen barrier ability, which dictates the applicability of the proposed packaging to enhance the quality of the packaged foods by retarding microbial growth and lowering lipid oxidation, thus extending their shelf-life [8,9,28]. The OP of the PLA film and films with a GC/NF coating, with and without BLEE, is presented in Table 1. The uncoated PLA film displayed the highest OP (*p* < 0.05), indicating its poor oxygen barrier ability. This might be related to the non-polar nature of the PLA, thereby allowing the diffusion of oxygen, which is known as a non-polar molecule [1,8]. However, when GC/NF was laminated on the PLA film by electrospinning, the OP decreased considerably (*p* < 0.05) and this reduction was augmented, when the concentration of BLEE increased (*p* < 0.05). A similar reduction in the OP was observed in film coated with nanospun fibers [8,9,29]. The reduction in OP could be related to the high polarity of the electrospun PLA films, induced by the addition of the GC/NF layer and further increased with the loading of BLEE that contains many polar compounds, thus posing very low oxygen permeation [8]. Additionally, Liu et al. [9] attributed a lowered OP to the high cross-linking reactions of the different polymers with PLA film, minimizing the free area available for oxygen molecules to diffuse through the film, and increasing the distance that the oxygen molecules must cross in the film due to the increased tortuosity.

#### 2.1.6. Color, Light Transmission, and Transparency

Color attributes of different films are tabulated in Table 1. Coating with GC/NF augmented the film cloudiness, which was in tandem with the lower L* values as compared with uncoated PLA films. The L* value continued to decrease, particularly with increasing BLEE concentrations (*p* < 0.05). Apart from the reduction in L*, the GC/NF-coated PLA films had an increased yellowish color when BLEE was encapsulated, particularly when BLEE at 1 and 2% was loaded into the GC/NF (*p* < 0.05). The films with the high BLEE concentration (1 and 2%) had the lowest a* value, representing more greenness. This might be caused by an amount of chlorophyll being retained in the BLEE. ∆E* value also increased with augmenting levels of BLEE in the GC/NF (*p* < 0.05), indicating the elevated change in color and the increases in both greenness and yellowness. The color of the PLA films was more likely attributed to some remaining pigmented compounds in the BLEE embedded in the GC/NF.

Light transmission in ultraviolet (UV) and visible (Vis) domains (200–800 nm) decreased significantly in all the GC/NF-coated PLA films, irrespective of BLEE loading (Table 2). An elevated UV/Vis light barrier ability was ascribed to the amino acid residues, especially those containing aromatic rings with double bonds in GC/NF, that can absorb UV light [30].

Furthermore, the loading of BLEE into GC/NF lowered the transmittance due to the diverse phenolic compounds or other components in BLEE, such as vitexin and isovitexin [12]. These compounds contain abundant aromatic rings and unsaturated bonds in their structures, which have the ability to absorb UV/Vis light [31]. Films with excellent UV/Vis light barrier abilities are highly regarded for the prevention of the adverse effects of light toward quality changes, especially lipid oxidation.

The transparency values of the different films (Table 2) showed that only the uncoated PLA film was transparent, while the coated counterparts were opaque. GC/NF-coated films, particularly those with GC/NF loaded with BLEE, had the highest transparency values, indicating a lower transparency in these films. These high transparency values coincided with the lower L* values and decreased transmission. Overall, the electrospinning of GC/NF containing BLEE on the surface of PLA films considerably reduced their transparency, thus serving as a UV/Vis light barrier and preventing its damaging effect on packaged foods.

#### 2.1.7. Bioactivities of PLA Films Coated with Electrospun GC/NF without and with BLEE at Different Concentrations

##### Antioxidant Activities (AO-A)

The term ‘active packaging’ describes films with bioactivities, such as antioxidant or antibacterial activities, which are able to prevent food deterioration by retarding lipid oxidation and inhibiting microbial growth [32]. GC/NF loaded with BLEE and coated on PLA films increased the film AO-A (*p* < 0.05) (Table 3). This increase was in a BLEE concentration-dependent manner. GC/NF-0%BLEE was noted to have very low DPPH and ABTS radical-scavenging activity. In addition, it had no ferric-reducing or metal-chelating abilities. This might be attributed to the absence of BLEE. Nevertheless, the AO-A of this particular sample was more likely linked to the presence of residual amino groups of chitosan and -NH_2_ or -OH groups in the side chains of gelatin that might contribute to radical-scavenging activities [33]. AO-A increased markedly with the augmenting concentration of BLEE added into GC/NF (*p* < 0.05). The highest AO-A tested by all assays was highest in GC/NF-2%BLEE, which possessed the highest level of BLEE (*p* < 0.05). This increase was strongly connected to the numerous polyphenols in the BLEE, such as isovitexin, epigallocatechin, and kaempferol derivatives, possessing high AO-A [12]. These compounds acted as electron donors or free-radical scavengers and exhibited the reduction of ferric ions into ferrous ions, and a metal-chelating ability [34]. Active packaging with such abilities, particularly those with reducing power and radical-scavenging activities, are highly regarded for their ability to lower lipid oxidation of foods and restrain the oxidative stress [35].

##### Antibacterial Activities (AB-A)

Various antibacterial activities of different films were observed against the tested bacteria (Table 3). Only the PLA films coated with GC/NF loaded with BLEE showed AB-A against all tested bacteria and the activity increased with augmenting concentration of BLEE (*p* < 0.05). Previous reports documented that the AB-A of the chitosan were related to its amino groups that could interact with microbial cell membranes, resulting in the leakage of intracellular proteinaceous constituents and the death of cells [36,37]. Lack of AB-A of the chitosan in the current study was probably caused by the strong fixation of chitosan in the coating layer and complexation with gelatin. This resulted in the weak diffusion of chitosan into the agar surface, so no AB-A was obtained. Wang et al. [38] also found no noticeable AB-A by pure chitosan films due to the firm fixation of chitosan and its poor diffusion into the media. Therefore, it could be presumed that most AB-A were related to the encapsulation of BLEE, which could diffuse into the agar surface and inhibited bacterial growth. The AB-A of BLEE were associated with the copious amounts of polyphenols and other components, such as essential oil, etc., within BLEE that had the profound ability to inhibit the bacterial growth [34,39]. Gram-positive bacteria (*S. aureus* and *L. monocytogenes*) were observed to show more resistance towards PLA films coated with GC/NF containing BLEE than Gram-negative bacteria (*E. coli* and *P. aeruginosa*). This could be owing to the presence of a thick layer of peptidoglycan in the Gram-positive bacteria, which showed more resistance against the actions of antibacterial agents [40]. Hydroxyl groups of polyphenols in BLEE plausibly acted as proton exchangers and destabilized the cytoplasmic membrane of the bacterial cells. This could further reduce the pH gradient and disturb the membrane ion permeability, thereby leading to the disruption of the electron flow and the proton motive forces. Those changes result in the discharge of ATPs and other constituents required for the metabolism of bacterial cells and, finally, cause bacterial death [39,41].

#### 2.1.8. Thermal Stability

The thermal degradation behavior of GC/NF-2%BLEE showing the most satisfactory mechanical, barrier, and bioactive properties in comparison with uncoated PLA film was evaluated by TGA (Figure 2A,B). Similar thermal degradation behavior was observed at the first stages of the analysis. The first significant weight loss (∆*w*) took place at an approximate thermal degradation temperature (T_d_) of 100 °C, where weight losses of 6.5 and 7.062% were observed for PLA and GC/NF-2%BLEE films, respectively. Such losses can be ascribed to the evaporation of moisture and residual solvent in the PLA film, and to the loss of volatile compounds in BLEE in the case of GC/NF-2%BLEE film, which showed a slight increase in its ∆*w* [14]. The maximum rate of degradation (∆*w* = 91.054 and 76.052%) occurred at 340.16 and 315.05 °C for PLA and GC/NF-2%BLEE films, respectively. PLA usually decomposes at temperatures ranging between 285 and 368 °C, in which cyclic oligomers, lactide molecules, CO, and CO_2_ are released [42]. The lower T_d_ of GC/NF-2%BLEE was probably due to the lower PLA polymerization and high degradation of the gelatin/chitosan nanofibers, causing the film to thermally degrade at a slightly higher rate. A similar degradation trend was reported by Boudjema, Bendaikha, and Maschke [43] who observed that PLA films coated with *Atriplex halimus* fibers had a lower T_d_ than that of the PLA film; this resulted from the irregularity of the fiber distribution on the PLA film with the presence of some agglomerations. Both film samples continued to degrade until the end of heating (800 °C). Nevertheless, the rate of degradation was much lower for GC/NF-2%BLEE film, as ascertained by the higher residue. This could be attributed to the thermal stability of several components in the nanofiber coating, indicating better thermal stability. The residual mass remained after completion of analysis was 0.132 and 6.972% for the PLA and GC/NF-2%BLEE films, respectively. In addition, the low degradation could be related to the affinity between the GC/NF-2%BLEE film components [43]. The derivative weight-loss curves of the films (Figure 2B) show another step process overlapping PLA decomposition, which might be linked to the GC/NF-2%BLEE coating. A similar behavior was observed by Arrieta et al. [44] when PLA films were loaded with limonene.

### 2.2. Quality Changes of Nile Tilapia Slices during Refrigerated Storage

#### 2.2.1. Changes in Microbiological Load

Bags made from GC/NF-2%BLEE, which showed the most satisfactory physical and bioactive properties, were used for packaging of tilapia slices. Changes in the microbiological load of tilapia slices packaged in these bags were compared to those packaged in normal LDPE bags (control). Bacterial counts were monitored during 12 days of refrigerated storage (Figure 3). At day 0, the total viable count (TVC) of slices packaged in either LDPE or GC/NF-2%BLEE bags (Figure 3A) ranged from 4.30 to 4.37 log CFU/g, indicating a slight contamination that probably took place during handling or preparation. A rapid increase in TVC was observed for both samples at day 3 (*p* < 0.05); however, the increase was much lower for the slices stored in the GC/NF-2%BLEE bags (5.53 log CFU/g) than for those packaged in the LDPE bags (6.55 log CFU/g). Since the accepted TVC marginal limit for most fresh water and marine fish species is 6.0 log CFU/g [45], it can be deduced that the LDPE bags could maintain the eating quality for only 3 days, while the GC/NF-2%BLEE bags showed a better preservative effect, more likely due to the BLEE diffusing from the GC/NF and acting as an antibacterial agent towards any spoilage bacteria. The TVC continued to increase markedly for the control (*p* < 0.05) until the end of the storage period, confirming the unsuitability of LDPE bags for tilapia slice preservation. On the other hand, the GC/NF-2%BLEE bags were able to maintain a TVC below the accepted limit after 6 days of storage (5.72 log CFU/g). At day 9, the TVC of the slices packaged in the GC/NF-2%BLEE bags was still below the limit (5.97 log CFU/g); however, the TVC surpassed the limit at day 12 (6.67 log CFU/g), denoting the ability of the GC/NF-2%BLEE bags to maintain a TVC within the acceptable limit for 9 days under refrigerated conditions.

The psychrophilic bacteria count (PBC) for tilapia slices packaged in the LDPE and GC/NF-2%BLEE bags was slightly increased within 3 days of storage (*p* < 0.05) (Figure 3B), with the counts of 3.65 and 3.38 log CFU/g, respectively. At day 6, the PBC for the slices packed in the GC/NF-2%BLEE bag was 5.54 log CFU/g, but the PBC of the slices packaged in the LDPE bag upsurged and reached 6.14 log CFU/g. During 9–12 days, the PBC kept increasing for both samples. Nevertheless, the rate of increase was much lower in the slices packaged in the GC/NF-2%BLEE bags, which had a PBC of 5.98 log CFU/g at day 9. Psychrophilic bacteria are known for their ability to flourish at low temperatures, thus causing spoilage of fish samples under refrigerated storage [34]. Therefore, retardation of psychrophilic bacteria growth was crucial to prevent or retard the undesired spoilage.

The majority of food spoilage, especially in fish and its products, was linked to the presence of a high load of *Pseudomonas* sp., known for their endurance and high survival rate, particularly at refrigerated temperatures [46]. The PC for both samples showed a marked increase after day 0 (Figure 3C), but the increase rate was greater in slices packaged in the LDPE bags, which had a PC higher than 6.0 log CFU/g after 9 days of storage, while the PC was lower for the slices packaged in the GC/NF-2%BLEE bags at the same time of storage. A lower PC rate of increase was observed for the slices kept in the GC/NF-2%BLEE bags throughout the storage period. The GC/NF-2%BLEE bags, therefore, had a superior ability in retarding the growth of *Pseudomonas* sp. in tilapia slices compared with the LDPE bags.

A similar trend was noticed for H_2_S-BC (Figure 3D), for which growth was accelerated in the slices packaged in LDPE, compared with slices kept in the GC/NF-2%BLEE bags throughout the storage period. The results signify the higher ability of GC/NF-2%BLEE bags in lowering the growth of H_2_S-producing bacteria, well known for their capability in utilizing amino acids, such as methionine and L-cysteine, besides other amino acids in proteins present in fish meat. This was associated with the generation of the unpleasant odor of H_2_S [47].

The presence of *Enterobacteriaceae* sp. is related to highly unhygienic or improper conditions for the preparation of fish [48]. Under suitable growth conditions, *Enterobacteriaceae* sp. counts can augment rapidly even if the initial count is very low [34]. Although EC was generally lower than the other bacterial counts (Figure 3E), it followed the same trend. Slices packaged in the GC/NF-2%BLEE bags generally showed a lower rate of increase compared with those kept in LDPE bags during the 12 days of storage. The result confirmed the appropriateness of the GC/NF-2%BLEE bags against the growth of a broad spectra of food spoilage bacteria mainly due to the highly diffusible BLEE from GC/NF coated on PLA film, thus acting as an antibacterial agent.

#### 2.2.2. Changes in Chemical Indices

Tilapia slices containing polyunsaturated fatty acids (PUFA) and other unsaturated fatty acids underwent lipid oxidation induced by free radicals and active oxygen species. This led to hydroperoxide formation along with other lipid oxidation products, which are responsible for the rancidity and the nutritional loss of fish [49]. Additionally, the contaminated bacteria can cause lipolysis, initiating the release of free fatty acids that could be further utilized by these bacteria as an energy source and in other biochemical processes [50]. These FFAs are highly prone to oxidation associated with the formation of hydroperoxides and other secondary lipid oxidation products, such as malonaldehyde and aldehydes [50]. At day 0, no difference in peroxide values (PV) could be detected between the slices packaged in the LDPE and GC/NF-2%BLEE bags (*p* > 0.05) (Figure 4A). An increase (*p* < 0.05) was observed at day 3 in both samples; however, the increase was lower in the slices packaged in the GC/NF-2%BLEE bags. The PV increment continued for both samples, but a lower rate was noticeable in the samples packaged in the GC/NF-2%BLEE bags. By day 9, the PV of the slices packaged in LDPE bags reached 9.23 mg cumene hydroperoxide/kg, which exceeded the accepted PV limit of 9 mg cumene hydroperoxide/kg [51]. The PV of the slices packaged in the GC/NF-2%BLEE bags was 7.01 mg cumene hydroperoxide/kg after the same period, denoting better preservative effects towards lipid oxidation. The result was in line with that of microbial growth retardation at the same period. After 9 days, the PV remained constant for the slices packaged in LDPE bags, signifying the slow formation of new hydroperoxides and the existing decomposition. Nevertheless, the PV of slices in GC/NF-2%BLEE bags continued to increase. By the end of storage (12 days), both samples surpassed the accepted limit and no differences in PV were detected (*p* > 0.05). Therefore, GC/NF-2%BLEE bags could prevent lipid oxidation of slices for 9 days under refrigerated conditions.

The TBARS value indicates the extent of lipid oxidation by the determination of any formed secondary oxidation products, such as ketones, aldehydes, etc., [52]. No difference in TBARS was observed up to day 3 (Figure 4B) (*p* < 0.05). At day 3, the slices packaged in the GC/NF-2%BLEE bags showed a slightly lower TBARS value (0.94 mg MDA/kg) than those packaged in the LDPE bags (1.22 mg MDA/kg). The TBARS values for both samples upsurged with increasing storage time but, in general, the increase was lower for slices packaged in the GC/NF-2%BLEE bags. Ozogul et al. [53] reported that TBARS values higher than 4 mg MDA/kg were considered as an indication of unacceptable quality. At day 9, the slices packaged in the LDPE bags had a TBARS value of 4.26 mg MDA/kg, while those kept in the GC/NF-2%BLEE bags had a value of 3.07 mg MDA/kg. The results indicate the capability of the latter to lower TBARS formation for 9 days. In addition to the antioxidant compounds from the BLEE embedded in GC/NF that were able to scavenge the free radicals causing the oxidation, gelatin and chitosan in the nanofiber matrix could act as an oxygen barrier, thus lowering the oxidation of lipid in the slices [54].

TVB content can reflect fish quality by quantifying the total nitrogenous compounds, such as ammonia (NH_3_), dimethylamine (DMA), and trimethylamine (TMA), formed due to the uncontrollable growth of contaminated microorganisms [55]. Usually, high TVB contents are connected to the poor quality of the fish and augmented levels are connected with an increase in the pH of tested samples, since most compounds generated are basic in pH, which could favor the growth of many spoilage bacteria [50]. No differences in TVB contents (6.82 and 6.37 mg N/100 g) or pH (6.06 and 6.07) for the slices packaged in the LDPE and GC/NF-2%BLEE bags were found at day 0 (*p* > 0.05) (Figure 4C,D). Both indices showed a marked increase (*p* < 0.05) for both samples with augmenting storage time, but the rate of increase was much higher (*p* < 0.05) for the slices packaged in LDPE bags. At day 6, the TVB content of the slices packaged in the LDPE bags was 39.60 mg N/100 g. TVB content higher than 35 mg N/100 g is regarded by the European Union (EU) Commission [56] as an indication of poor-quality fish. It can be inferred that the LDPE bags could maintain the slice quality for no longer than 6 days. On the other hand, the slices packaged in the GC/NF-2%BLEE bags had a TVB content of 33.69 mg N/100 g after 9 days of storage, signifying its acceptable quality. However, the TVB content after 12 days surpassed the accepted limit of freshness. Thus, the preservative ability of the GC/NF-2%BLEE bags in maintaining the quality of tilapia slices for 9 days was mainly due to the presence of BLEE which was able to retard microbial growth. The increase in the TVB contents of both samples was accompanied with a coincidental increase in pH. Nevertheless, after 12 days of storage, the pH of the slices packaged in the GC/NF-2%BLEE bags was 6.83, while that of the slices packaged in the LDPE bags was 7.98. The result confirmed the improved keeping quality of the tilapia slices by the GC/NF-2%BLEE bags.

## 3. Materials and Methods

### 3.1. Materials

Fish gelatin powder (bloom: 250; molecular weight (MW): ~5.1 × 10^4^ Da) was acquired from Lapi Gelatine S.P.A (Empoli, Italy). Chitosan (MW: ~2.1× 10^3^ KDa; degree of deacetylation (DDA): ~82%) was purchased from Sigma-Aldrich (St. Louis, MO, USA). Acetic acid and ethanol were procured from RCI Labscan Limited (Bangkok, Thailand). Polylactic acid pellets were obtained from Nature Work Co. Ltd. (Blair, NE, USA).

### 3.2. Preparation and Dechlorophyllization of Betel Leaf Ethanolic Extracts (BLEE)

Extraction and dechlorophyllization of BLEE were performed as described by Tagrida and Benjakul [12]. Betel leaf powder was mixed with ethanol (70%) at 1:15 (*w*/*v*) and the mixture was stirred for 60 min. After filtration using filter paper, the solvent was removed from the filtrate with the aid of a rotary evaporator. The sedimentation method was applied to the concentrated extract for dechlorophyllization, in which the extract was mixed with distilled water at a 1:1 ratio and left at 4 °C for 24 h. Thereafter, the dechlorophyllized extract was freeze dried, and the powder was stored in a capped vial at −20 °C until use.

### 3.3. Preparation of Polylactic Acid (PLA) Film

A casting technique was adopted for the PLA film preparation. PLA pellets were mixed with chloroform to obtain a final concentration of 5% (*w*/*v*). Glycerol (1%, *w*/*v* based on solid content) was added as a plasticizer. The mixture was stirred using a magnetic stirrer at room temperature until complete solubilization was achieved. The PLA solution (200 mL) was degassed for 10 min in a sonication bath. Subsequently, the solution was poured into a stainless-steel tray (35 × 25 cm^2^) and placed for 3 days at room temperature in an aerated chamber for solvent evaporation. PLA films were peeled from the trays and conditioned in an environmental chamber (Binder GmbH, Tuttlingen, Germany) with a relative humidity (RH) of 50 ± 5% and a temperature of 25 ± 0.5 °C for 48 h before analyses.

### 3.4. Preparation of Electrospun Gelatin/Chitosan Nanofiber (GC/NF) Incorporated with BLEE

Fish gelatin was added to 90% (*v*/*v*) acetic acid to obtain 20% concentration (*w*/*v*) and then stirred at 250 rpm using a magnetic stirrer for 2 h at 45°C until complete dissolution. Thereafter, chitosan was mixed with gelatin solution at 3% (*w*/*v*) and stirred constantly until the chitosan was totally solubilized. BLEE was added into a 10 mL gelatin/chitosan (GC) mixture to obtain several concentrations (0.5, 1, and 2% (*w*/*v*)). The solutions, namely GC-0.5%BLEE, GC-1%BLEE, and GC-2%BLEE, were further used as the electrospinning solutions. GC mixture devoid of BLEE was used as the electrospinning solution and served as the control.

The electrospinning process was conducted at a temperature of 25 ± 2 °C and 45 ± 2% RH as specified by Gulzar, Tagrida, Nilsuwan, Prodpran, and Benjakul [18]; the process was performed by injecting the solutions into a plastic syringe equipped with a stainless-steel needle (23-guage). The syringe was then loaded to the pump of the electrospinning machine (NanoSpinner, Inovenso Technology Inc., Woborn, MA, USA). The needle was positioned 12 cm away from the stainless-steel drum collector (10 cm diameter and 20 cm length) which was adjusted at a rotation rate of 200 rpm and oscillation of 10 mm/s. A previously prepared PLA film (12 × 30 cm^2^) was wrapped around the drum collector and the electrospinning solution was pumped at a feed rate of 0.4 mL/h with 25 kV applied voltage. A collection time of 24 ± 1 h was used. The obtained films were then removed from the drum collector and further conditioned in an environmental chamber for 24 h at a temperature of 25 ± 0.5 °C and 50 ± 5% RH until analysis. Films were named GC/NF-0%BLEE, GC/NF-0.5%BLEE, GC/NF-1%BLEE, and GC/NF-2%BLEE for PLA film samples coated with GC/NF and loaded without BLEE (0%), and with BLEE at 0.5, 1, and 2%, respectively.

### 3.5. Characterization of PLA Films Coated with Electrospun GC/NF without and with BLEE at Various Concentrations

#### 3.5.1. Morphology

The microstructure of nanofibers was analyzed by a scanning electron microscope (SEM) (Quanta 400, FEI, Eindhoven, the Netherlands). A gold layer was sputter-coated on the film samples using a sputter coater (SPI-Module, West Chester, PA, USA). Thereafter, films were visualized at an acceleration voltage of 20 kV.

#### 3.5.2. Thickness

The thickness of films was determined using a digital micrometer (Mitutoyo, Model ID-C112PM, Mitutoyo Corp., Kawasaki-shi, Japan). Nine random positions on the film samples were measured for thickness and the average thickness was calculated.

#### 3.5.3. Mechanical Properties

Tensile strength (TS) and elongation at break (EAB) were determined as specified by Nilsuwan et al. [1]. TS (MPa) was computed by dividing the maximum force (N) required to break down film with a designated cross-section area (m^2^). EAB was computed by dividing the elongation of the film with the initial grip length of the film samples (30 mm). The crosshead speed was fixed at 30 mm/min and the EAB was reported as a percentage.

#### 3.5.4. Water Vapor Permeability (WVP)

The method described by Nilsuwan et al. [1] was followed. Film samples were placed on the aluminum permeation cups containing 0% RH dried silica gel (20 g), in which the film coated with nanofibers was facing the silica gel. Thereafter, the films were tightly sealed on the cups using silicon vacuum grease and a rubber gasket. The cups were kept in an environmental chamber (25 ± 2 °C and 50 ± 5% RH) and were weighed every 1 h over a 10-h period. WVP was subsequently computed as follows:(1)$$\mathrm{WVP}\;(g\;m^{-1}\;s^{-1}\;{\mathrm{Pa}}^{-1})=(\frac{\mathrm{WVTR}\;\times \;L}{\Delta P\;\times \;A})$$
where WVTR is the water vapor transmission rate obtained from the slope of plot between cup weight gain vs time (g/s), L is thickness of the film (m), ∆P is the water vapor pressure difference across the film (1588.69 Pa at 25 °C), and A is the exposed area of the film (m^2^).

#### 3.5.5. Oxygen Permeability (OP)

The OP of the prepared films was determined as specified by Nilsuwan et al. [1]. The oxygen transmission rate (OTR) through each film was measured using an oxygen permeation analyzer (model 8000, Illinois Instruments Inc., Johnsburg, IL, USA). Films were clamped in the diffusion chamber and pure oxygen (99.8% purity) was then introduced into the upper half of the sample chamber, while nitrogen was injected into the lower half of the chamber where there was an oxygen sensor. The OTR was determined at 25 °C and 50% RH. Measurements were performed in triplicate. OP was calculated using following equation:(2)$$\mathrm{OP}\;(\mathrm{mol}\;m^{-1}\;s^{-1}\;{\mathrm{Pa}}^{-1})=\frac{\mathrm{OTR}\;\times \;L}{\Delta P}$$
where OTR is the oxygen transmission rate (mol/m^2^.s), L is thickness of the film (m), and ∆P is the partial pressure of oxygen (1.015 × 10^5^ Pa at 25 °C).

#### 3.5.6. Color, Light Transmission, and Transparency

The colors of the films were determined as reported by Mittal et al. [36] using a colorimeter (Hunterlab, Reston, VA, USA). L* (lightness), a* (redness/greenness), b* (yellowness/blueness), and ΔE* (total color difference) were recorded.

The method of Shiku, Hamaguchi, Benjakul, Visessanguan, and Tanaka [57] was adopted for the determination of light transmission of the films in both ultraviolet (200–280 nm) and visible (350–800 nm) ranges with the aid of an UV-vis spectrophotometer (UV-1800, Shimadzu, Kyoto, Japan). The transparency value was computed [18]:(3)$$\mathrm{Transparency}\;\mathrm{value}=\frac{-\log T_{600}}{X}$$
where T_600_ is the fractional transmission at 600 nm and X is film thickness (mm). A lower film transparency is indicated by a higher transparency value.

#### 3.5.7. Bioactivities of PLA Films Coated with Electrospun GC/NF without and with BLEE at Various Concentrations

##### Antioxidant Activities (AO-A)

Small pieces of film (0.1 g) were mixed with distilled water (10 mL) and stirred overnight. The mixture was centrifuged for 20 min at 8000× *g*, and the supernatants were collected and used for determination of AO-A following the method of Tagrida, Prodpran, Zhang, Aluko, and Benjakul [58]. DPPH and ABTS radical-scavenging activities (DPPH-RA and ABTS-RA, respectively), and ferric-reducing antioxidant power (FRAP) were examined and reported as μmol Trolox equivalent (TE)/g sample. Metal-chelating activity (MCA) was expressed as μmol EDTA equivalent (EE)/g sample.

##### Antibacterial Activity (AB-A)

AB-A of different films was assessed by the agar diffusion method described by Giménez et al. [11]. *Staphylococcus aureus*, *Escherichia coli*, *Listeria monocytogenes*, and *Pseudomonas aeruginosa* gifted from the Food Safety Laboratory, Prince of Songkla University, Hat Yai, Thailand, were the tested microorganisms. One hundred μL of bacterial suspension (10^6^ CFU/mL) was spread on tryptic soy agar (TSA) plates under aseptic conditions. Subsequently, film samples cut into discs (diameter: 5 mm) were placed on the surface of the inoculated TSA plates. All the plates were incubated for 24 h at 37 °C. The inhibition zone dimeter including the film sample was then determined in millimeters.

#### 3.5.8. Thermogravimetric Analysis (TGA)

The thermostability of the selected film was evaluated with the aid of a simultaneous thermal analyzer (STA-8000, PerkinElmer, Norwalk, CT, USA) using a heating flow rate of 10 °C/min from 30 to 800 °C. The thermal degradation temperature and weight loss of the film samples were recorded. Nitrogen was used as the purge gas at a flow rate of 20 mL/min.

### 3.6. Quality Changes in Nile Tilapia Slices Packaged in Bags Prepared from the Selected PLA Films Coated with GC/NF Containing BLEE during Refrigerated Storage

#### 3.6.1. Preparation of Nile Tilapia slices

Freshly deceased Nile tilapia (3 kg; 1 ± 0.1 kg per fish) were bought from the local market and the fish were placed in ice (1:2 ratio). Thereafter, the fish were transported to the laboratory within 30 min where fish slices were prepared as per the method of Tagrida, Benjakul, and Zhang [34]: the fish were washed with tap water and decapitated, skinned, filleted, and sliced (4 × 2 cm^2^) using a stainless-steel knife. Slices were kept in ice during preparation.

#### 3.6.2. Preparation of Bags

Selected film samples (28 × 7 cm^2^) were folded in half, in which the GC/NF coated side was the inner side of the bags and then thermally sealed on 2 sides (150 ± 0.5 °C for 5 s, followed by cooling for 1.50 s) with the aid of an impulse sealer (magnet model ME-300HIM, S.N. Mark Ltd., Park, Nonthaburi, Thailand). The width of the seal area was 2 mm, and the final dimensions of the bags were 14 × 7 cm^2^. Fish slices (10 g) were placed in the prepared bags, which were subsequently closed via sealing. Fish slices were placed in LDPE bags with the same dimensions and sealed to serve as the control. All bags were stored at 4 °C and quality assessments were conducted on randomly taken samples every 3 days up to 15 days.

#### 3.6.3. Microbiological Analyses

The spread plate method detailed by Tagrida and Benjakul [59] was adopted for monitoring the changes in the microbiological load of fish slices during storage. A volume of 45 mL of 0.85% sterilized saline solution was added to slices (5 g) in a stomacher bag. The mixture was mixed well using a stomacher blender (M400, Seward Ltd., West Sussex, England) for 1 min at 220 rpm. The required serial dilutions were performed using 0.85% sterilized saline solution and 0.1 mL of the appropriate dilutions were spread on the surface of the corresponding media for enumeration. Plate count agar was used for the enumeration of total viable count (TVC) and psychrophilic bacterial count (PBC) after incubation at 37 °C for 3 days and 4 °C for 10 days, respectively. Triple sugar iron agar and Pseudomonas isolation agar were used for the H_2_S-producing bacterial count (H_2_S-BC) and the *Pseudomonas* sp. count (PC), respectively. The counts were recorded after incubation for 3 days at 25 °C. The *Enterobacteriaceae* sp. count (EC) was evaluated after inoculation on eosin methylene blue agar and incubation at 37 °C for 24 h.

#### 3.6.4. Chemical Analyses

Lipid oxidation of slices was evaluated by determination of the peroxide value (PV) as specified by Richards and Hultin [60] and was expressed as an mg cumene hydroperoxide/kg sample. Thiobarbituric acid reactive substances (TBARS) values were examined as described by Buege and Aust [61] and expressed as an mg malonaldehyde (MDA)/kg sample. Total volatile base (TVB) content was measured following the method of Olatunde, Benjakul, and Vongkamjan [45] and reported as an mg N/100 g sample. The pH of the sample homogenate (slices: distilled water = 1:10, *w*/*v*, 13,000 rpm, for 1 min) was measured using a pH-meter (Sartorious North America, Edgewood, NY, USA).

### 3.7. Statistical Analysis

Completely randomized design (CRD) was applied for all studies. Data were reported as mean ± standard deviation (SD). For mean comparison, One-way Analysis of Variance (ANOVA) was conducted, and Duncan’s Multiple Range Test (DMRT) was performed. SPSS package (SPSS 23.0 for Windows, SPSS Inc, Chicago, IL, USA) software was used for data analysis.

## 4. Conclusions

Electrospun gelatin/chitosan nanofibers (GC/NF) with and without betel leaf ethanolic extract (BLEE) were coated on PLA films. The nanofibers appeared to be bead-free, with a fibrous network structure particularly at low concentrations of BLEE. Mechanical and barrier properties were improved by the nanofiber coating. The WVP was lowered with an increasing BLEE concentration, while the tensile strength and stretch ability of PLA film coated with GC/NF were considerably enhanced, particularly when BLEE at 1 and 2% was added in GC/NF. BLEE loaded into GC/NF, especially at higher concentrations, provided antioxidant and antibacterial activities to PLA films. Bags prepared from GC/NF-coated PLA films and loaded with 2% BLEE (GC/NF-2%BLEE) could impede microbial growth in tilapia slices as compared with LDPE bags under refrigerated storage. Additionally, GC/NF-2%BLEE bags prevented lipid oxidation and other chemical changes associated with microbial spoilage in slices during storage. Therefore, PLA films coated with GC/NF containing BLEE could serve as a promising active packaging for food preservation.

## Figures and Tables

**Figure 1 molecules-27-05877-f001:**
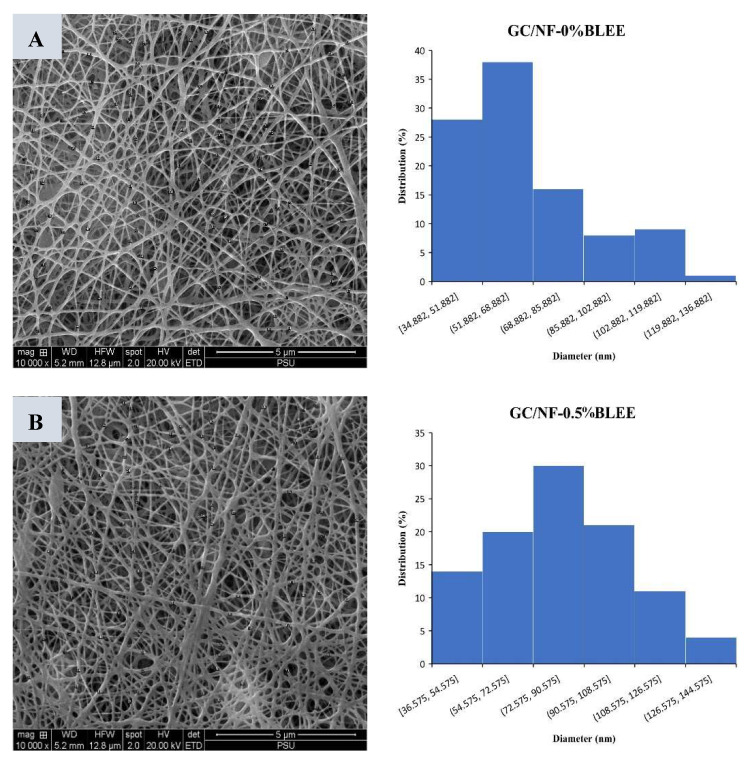
Scanning electron microscopic (SEM) images and histograms representing nanofiber diameter distributions coated on the PLA films containing betel leaf ethanolic extract (BLEE) at 0% (**A**), 0.5% (**B**), 1% (**C**), and 2% (**D**).

**Figure 2 molecules-27-05877-f002:**
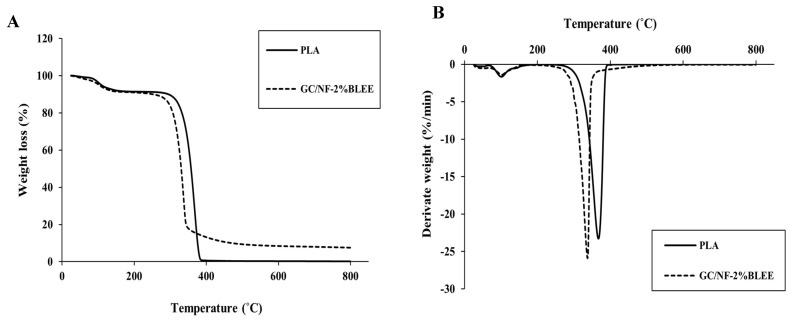
Thermogravimetric analysis (**A**) and derivative weight loss (**B**) thermograms of GC/NF-2%BLEE film and PLA film.

**Figure 3 molecules-27-05877-f003:**
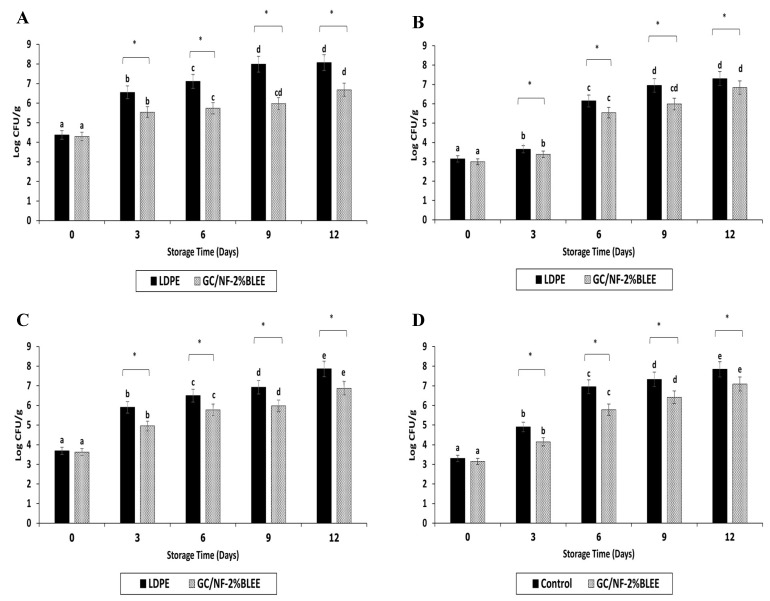
Total viable count (**A**), psychrophilic bacteria count (**B**), *Pseudomonas* sp. count (**C**), hydrogen sulfide-producing bacteria count (**D**), and *Enterobacteriaceae* sp. count (**E**) of tilapia slices packaged in different bags, during 12 days of refrigerated storage. Bars represent standard deviation (n = 3). Single asterisk (*) indicates significant difference (*p* < 0.05). Different lowercase letters of samples packaged in the same bag indicate significant difference (*p* < 0.05).

**Figure 4 molecules-27-05877-f004:**
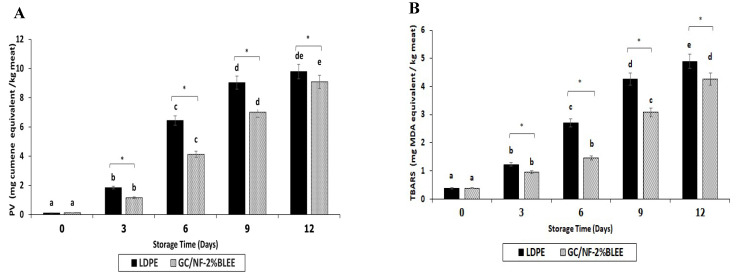
Peroxide value (PV) (**A**), thiobarbituric acid reactive substances (TBARS) (**B**), total volatile base (TVB) content (**C**), and pH (**D**) of tilapia slices packaged in different bags during 15 days of refrigerated storage. Bars represent standard deviation (n = 3). Single asterisk (*) indicates significant difference (*p* < 0.05). Different lowercase letters of samples packaged in the same bag indicate significant difference (*p* < 0.05).

**Table 1 molecules-27-05877-t001:** Thickness, mechanical properties, color, oxygen permeability, and water vapor permeability of PLA film and PLA films coated with GC/NF loaded with, or without betel leaf ethanolic extract (BLEE) at different levels.

Sample	Thickness (mm)	TS (MPa)	EAB (%)	L*	a*	b*	∆E	WVP (×10^−11^ g m^−1^ s^−1^ Pa^−1^)	OP (×10^−18^ mol m^−1^ s^−1^ Pa^−1^)
PLA	0.106 ± 0.004 ^a^	29.55 ± 0.54 ^a^	4.85 ± 1.00 ^a^	91.79 ± 0.01 ^d^	−0.01 ± 0.01 ^d^	0.07 ± 0.01 ^a^	-	4.10 ± 0.18 ^c^	216.6 ± 11.07 ^e^
GC/NF-0%BLEE	0.115 ± 0.001 ^b^	32.88 ± 1.50 ^b^	5.00 ± 0.97 ^a^	89.45 ± 1.05 ^cd^	−1.28 ± 0.06 ^c^	1.50 ± 0.20 ^a^	3.08 ± 0.84 ^a^	3.86 ± 0.20 ^bc^	25.88 ± 0.28 ^d^
GC/NF-0.5%BLEE	0.118 ± 0.001 ^bc^	34.03 ± 1.47 ^b^	5.45 ± 1.02 ^ab^	87.79 ± 1.19 ^bc^	−4.63 ± 0.51 ^b^	5.71 ± 0.46 ^a^	8.37 ± 0.82 ^b^	3.39 ± 0.20 ^b^	22.59 ± 2.07 ^c^
GC/NF-1%BLEE	0.121 ± 0.001 ^c^	34.40 ± 1.58 ^b^	6.80 ± 0.18 ^b^	85.60 ± 1.71 ^b^	−7.03 ± 0.51 ^a^	17.65 ± 1.04 ^b^	20.48 ± 1.95 ^c^	3.15 ± 0.21 ^ab^	10.54 ± 1.78 ^b^
GC/NF-2%BLEE	0.121 ± 0.002 ^c^	34.45 ± 2.65 ^b^	7.04 ± 2.26 ^b^	81.99 ± 3.02 ^a^	−7.43 ± 0.74 ^a^	20.20 ± 0.89 ^b^	23.78 ± 0.61 ^c^	3.00 ± 0.01 ^a^	6.23 ± 0.59 ^a^

Values are mean ± SD. Different lowercase superscripts in the same column denote significant difference (*p* < 0.05). TS: tensile strength; EAB: elongation at break; L*: lightness; a*: redness/greenness; b*: blueness/yellowness; WVP: water vapor permeability; OP: oxygen permeability. PLA: polylactic acid film without electrospinning. GC/NF-0%BLEE, GC/NF-0.5%BLEE, GC/NF-1%BLEE, and GC/NF-2%BLEE represent PLA films coated with GC/NF without BLEE (0%) and loaded with BLEE at 0.5, 1, and 2% BLEE, respectively.

**Table 2 molecules-27-05877-t002:** Light transmission and transparency values of PLA film and PLA films coated with GC/NF loaded with, and without, BLEE at different levels.

Sample	Light Transmission (%) at Different Wavenumbers (nm)								Transparency Value
	200	280	350	400	500	600	700	800	
PLA	0.06	49.75	69.33	78.79	83.80	88.21	88.48	88.60	0.51 ± 0.14 ^a^
GC/NF-0%BLEE	0.07	0.09	3.82	5.55	6.42	7.02	7.50	7.91	10.0 ± 0.33 ^b^
GC/NF-0.5%BLEE	0.06	0.06	2.05	3.03	3.93	4.87	5.05	5.28	11.05 ± 0.35 ^c^
GC/NF-1%BLEE	0.07	0.07	2.07	2.80	3.70	4.47	4.77	4.87	11.15 ± 0.23 ^c^
GC/NF-2%BLEE	0.07	0.09	1.91	2.61	3.67	4.38	4.63	4.71	11.21 ± 0.25 ^c^

Values are mean ± SD. Different lowercase superscripts in the same column denote significant difference (*p* < 0.05). For caption, see Table 1.

**Table 3 molecules-27-05877-t003:** Antioxidant and antibacterial activities of PLA film and PLA films coated with GC/NF loaded with and without BLEE at different levels.

Sample	Antioxidant Activities				Antibacterial Activity (mm)			
	DPPH-RSA (µmol TE/g sample)	ABTS-RSA (µmol TE/g sample)	FRAP (µmol TE/g sample)	MCA (µmol EDTA/g sample)	*S. aureus*	*E. coli*	*L. monocytogenes*	*P. aeruginosa*
**PLA**	ND *	ND	ND	ND	ND	ND	ND	ND
**GC/NF-0%BLEE**	3.50 ± 1.72 ^a^	25.55 ± 2.22 ^a^	ND	ND	ND	ND	ND	ND
**GC/NF-0.5%BLEE**	23.21 ± 0.69 ^b^	792.5 ± 8.41^b^	336.66 ± 5.50 ^a^	76.52 ± 3.91 ^a^	5.58 ± 0.50 ^a^	6.83 ± 0.57 ^a^	5.91 ± 0.68 ^a^	7.16 ± 0.88 ^a^
**GC/NF-1%BLEE**	75.89 ± 1.20 ^c^	1829.2 ± 7.22 ^c^	928.69 ± 8.72 ^b^	170.61 ± 1.13 ^b^	7.75 ± 0.63 ^b^	9.08 ± 0.31 ^b^	7.66 ± 0.72 ^b^	8.91 ± 0.87 ^b^
**GC/NF-2%BLEE**	93.49 ± 1.01^d^	2817.03 ± 7.39 ^d^	1484.3 ± 3.13 ^c^	277.37 ± 0.92 ^c^	9.08 ± 0.68 ^c^	11.25 ± 0.31 ^c^	9.33 ± 0.89 ^c^	10.66 ± 0.90 ^c^

Values are mean ± SD (*n* = 3). Different lowercase superscripts in the same column denote significant difference (*p* < 0.05). DPPH-RSA: 2,2-diphenyl-1-picrylhydrazyl radical-scavenging activity; ABTS-RSA: 2, 2’-azino-Bis-3-ethylbenzothiazoline-6-sulfonic acid radical-scavenging activity; FRAP: ferric-reducing antioxidant power; MCA: metal-chelating activity. For caption see Table 1. * ND: not detected.

## Data Availability

Data are not shared.

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
