# Peer review of "Polylactic Acid Film Coated with Electrospun Gelatin/Chitosan Nanofibers Containing Betel Leaf Ethanolic Extract: Properties, Bioactivities, and Use for Shelf-Life Extension of Tilapia Slices"

_molecules, 2022, doi:10.3390/molecules27185877_

Round 1

Reviewer 1 Report

The present study deals with "the fabrication of PLA films coated with electro-spun gelatin/chitosan nanofibers incorporated with betel leaf ethanolic extract (BLEE)". To this aim, the mechanical, physical, and bioactive properties of the resulting films were examined. The quality changes of tilapia slices stored in bags made from the selected PLA film coated with gelatin/chitosan electrospun fiber containing BLEE were also monitored during refrigerated storage.

In general, the manuscript has been well written. However, corrections/modifications, as well as some clarifications, should be considered. 

To assist the authors to improve the manuscript, I would like to give some suggestions and all the concerns are listed as follows:

Line 40-42: As the authors rightly mentioned, the commercial scale application of PLA has been limited due to its low thermal stability and gas barrier dysfunctionality. So, different approaches have been trialed in an attempt to enhance the gas barrier attributes of PLA. 

As an important analysis regarding this, the thermal stability and oxygen permeability of developed PLA films in the present study MUST be added to the manuscript. 

Meanwhile, GC/MS analysis of betel leaf ethanolic extract is required. 

Line 43: Since in the present study, the lamination technique was used to develop PLA-based film, the authors need to point out the importance of this technique.

Line 46: Please correct as follow: lamination of gelatin and chitosan on the PLA films can be overcome…

Line 47: It is recommended to the authors use the current research paper regarding the lamination of chitosan/gelatin/PVA to improve the barrier properties of PLA-based films. “Facile fabrication of transparent high-barrier poly(lactic acid)-based bilayer films with antioxidant/antimicrobial performances”

Line 80: Fiber diameter should be determined from 100 random measurements per image by using Image J software. Please provide the histogram of fiber diameter distributions for all images and added them to the manuscript. 

Line 82: Same as the previous comment. 

Line 82-85: The authors attributed the to the declined viscosity

Line 203-204: As the authors are aware, transparency can directly affect food appearance and consumer willingness and is an important criterion for packaging materials. In the present study, GC/NF coated films are completely opaque and the transparency values for the bilayer films were markedly lower than the reported values for LDPE (4.26 AU/mm), as the conventional plastic packaging. This means that the consumer cannot see the contents of the package, and therefore, they cannot judge the quality of the packaged product. Do you think there is a way to improve the transparency of these types of films? Which types of food products are suitable for packaging with these types of films? 

Line 225-227: GC/MS analysis of betel leaf ethanolic extract is required and MUST be added to the manuscript. This helps the authors to provide a more accurate interpretation of the antioxidant/antimicrobial results of the films.

Author Response

Response to reviewer

The present study deals with "the fabrication of PLA films coated with electro-spun gelatin/chitosan nanofibers incorporated with betel leaf ethanolic extract (BLEE)". To this aim, the mechanical, physical, and bioactive properties of the resulting films were examined. The quality changes of tilapia slices stored in bags made from the selected PLA film coated with gelatin/chitosan electrospun fiber containing BLEE were also monitored during refrigerated storage.

In general, the manuscript has been well written. However, corrections/modifications, as well as some clarifications, should be considered. 

To assist the authors to improve the manuscript, I would like to give some suggestions and all the concerns are listed as follows:

*****Thank you so much for the invaluable comments and suggestions. All queries have been responded and the corrections have been made as highlighted in yellow.

Line 40-42: As the authors rightly mentioned, the commercial scale application of PLA has been limited due to its low thermal stability and gas barrier dysfunctionality. So, different approaches have been trialed in an attempt to enhance the gas barrier attributes of PLA. 

As an important analysis regarding this, the thermal stability and oxygen permeability of developed PLA films in the present study MUST be added to the manuscript.

***** Thank you very much for valuable suggestions for elevating the level of our manuscript. All the required analyses were conducted, and the results and discussions were added. Please see lines 195 – 213 and 558-569 for oxygen permeability and line 305 – 333 and 601-606 for thermal stability.

Meanwhile, GC/MS analysis of betel leaf ethanolic extract is required. 

*****Betel leaf ethanolic extract was already analyzed in our previous work Tagrida and Benjakul (2020). The most abundant compounds of the extract were identified and provided in the text. Isovitexin was the most abundant compound followed by vitexin 4′-O-galactoside. The extraction protocol and material used was from the same lot. Thus, the compounds were not much varied. Therefore, there Some information on the aforementioned compounds of the extract was included in the current manuscript (lines 61-64).   

References

Tagrida, M., & Benjakul, S. (2020). Ethanolic extract of Betel (Piper betle L.) and Chaphlu (Piper sarmentosum Roxb.) dechlorophyllized using sedimentation process: Production, characteristics, and antioxidant activities. Journal of Food Biochemistry, 44(12), e13508. https://doi.org/10.1111/jfbc.13508

Line 43: Since in the present study, the lamination technique was used to develop PLA-based film, the authors need to point out the importance of this technique.

*****The mentioned issue has been addressed and the importance of the technique has been pointed out. Please see lines 42 - 46.  

Line 46: Please correct as follow: lamination of gelatin and chitosan on the PLA films can be overcome…

*****The required corrections have been made. Please see lines 49 - 50.

Line 47: It is recommended to the authors use the current research paper regarding the lamination of chitosan/gelatin/PVA to improve the barrier properties of PLA-based films. “Facile fabrication of transparent high-barrier poly(lactic acid)-based bilayer films with antioxidant/antimicrobial performances”

*****The required modifications have been added and the given research has been cited. Please see lines 51- 52.

Line 80: Fiber diameter should be determined from 100 random measurements per image by using Image J software. Please provide the histogram of fiber diameter distributions for all images and added them to the manuscript. 

Line 82: Same as the previous comment. 

*****The nanofibers diameters were determined using the mentioned software and the histograms of fiber diameter distributions were provided for each sample. Please see Figure 1.

Line 82-85: The authors attributed the to the declined viscosity

*****We apologize for this unintended ambiguity. The effect we meant in the manuscript is bead formation during the process of nanofiber electrospinning. Bead formation can take place due to many reasons. The most important reason was the declined viscosity of the nanofiber forming solutions. This resulted from the addition of the plant extract, and the change in the conductivity of the solutions during the electrospinning process. Possible reasons of the bead formation had been already discussed in more detail in the manuscript. Please see lines 94 – 116. For better clarification we added “Bead formation” to the text, please see line 91.

Line 203-204: As the authors are aware, transparency can directly affect food appearance and consumer willingness and is an important criterion for packaging materials. In the present study, GC/NF coated films are completely opaque and the transparency values for the bilayer films were markedly lower than the reported values for LDPE (4.26 AU/mm), as the conventional plastic packaging. This means that the consumer cannot see the contents of the package, and therefore, they cannot judge the quality of the packaged product. Do you think there is a way to improve the transparency of these types of films? Which types of food products are suitable for packaging with these types of films?

******We do agree that the transparency of the resulted films was very low. This is because of the electrospinning of gelatin/chitosan nanofibers on the PLA films. The nanofiber solutions were transparent before the process but when the high voltage was applied, the gelatin/chitosan jet was emanated and electro-spun on the PLA film as nanofibers mat. Unfortunately, we cannot control the transparency of the resulted nanofiber mat. However, when the bags made from such films were packaged with foods having high moisture content such as tilapia slices, the coated layer were partially dissolved, and the bags became transparent again. Thus, their quality appearance can be judged by the consumers. Picture of packaged tilapia slices in PLA film bags coated with gelatin/chitosan nanofibers containing the betel extract was provided in the graphical abstract.

The produced films are considered as active packaging because of its coating layer containing bioactive compounds (GC/NF containing BLEE). Therefore, the foods fitting well with such packages are those prone to microbial spoilage or lipid oxidation such as fish or fish products. Nevertheless, such packages may have the ability to pack several types of foods such as meat products, dried fruits, and nuts, etc.      

Line 225-227: GC/MS analysis of betel leaf ethanolic extract is required and MUST be added to the manuscript. This helps the authors to provide a more accurate interpretation of the antioxidant/antimicrobial results of the films.

******The same query was raised by the reviewer above. Please see the response on comment number 2.

Reviewer 2 Report

The paper submitted by Mohamed Tagrida et al. reported the effects of betel leaf ethanolic extract on the mechanical and barrier properties bioactivities of the polylactic acid film coated with electro-spun gelatin/chitosan. The research is interesting. This paper can be published in the journal after major revisions.

1. Please mention the novelty of the current study. The introduction part should be more informative with some updated references on polyester-based composites. Cite the following papers

a.       Composites Part B: Engineering, Volume 217, 2021, 108878.

b.      Food Packaging and Shelf Life, Volume 22, 2019, 100387

2. If possible the authors need to provide a typical S-S curve (tensile test) of fabricated samples.

3. To improve the paper's quality, the author should provide an oxygen barrier test of the composites.

4. Figures 2 and 3 need to present in better way (quality should be improved).

Author Response

Response to reviewer

The paper submitted by Mohamed Tagrida et al. reported the effects of betel leaf ethanolic extract on the mechanical and barrier properties bioactivities of the polylactic acid film coated with electro-spun gelatin/chitosan. The research is interesting. This paper can be published in the journal after major revisions.

*****Thank you so much for the invaluable comments and suggestions. All queries have been responded and the corrections have been made as highlighted in green.

  1. Please mention the novelty of the current study. The introduction part should be more informative with some updated references on polyester-based composites. Cite the following papers
  2. Composites Part B: Engineering, Volume 217, 2021, 108878.
  3. Food Packaging and Shelf Life, Volume 22, 2019, 100387

*****The novelty of the study was discussed, and the given references were cited. Please see lines 37, 69 – 71.

  1. If possible the authors need to provide a typical S-S curve (tensile test) of fabricated samples.

*****We do agree that such curves may add more clarity to the results. However, the results of the tensile test reported as numerical values was sufficient since many previous studies on biodegradable films and their characteristics also reported the tensile results as the values only. Therefore, we did not prefer to provide those curves. Sorry for this.

  1. To improve the paper's quality, the author should provide an oxygen barrier test of the composites.

*****The required test was performed, and the results were reported and discussed. Please see lines 195 – 212 and 558-569.

  1. Figures 2 and 3 need to present in better way (quality should be improved).

*****We apologize for this, those figures are of the best quality we could prepare, we could not increase the resolution of the figures more than 300 dpi. We are sorry for this again. 

Round 2

Reviewer 1 Report

The manuscript entitled “Polylactic acid film coated with electro-spun gelatin/chitosan nanofibers containing betel leaf ethanolic extract: Properties,
bioactivities, and its use for shelf-life extension of tilapia slices” has been well revised by authors and may be considered for publication in Molecules.

Reviewer 2 Report

The authors have addressed all the major concerns, so I would recommend it for publication.